# Detection of Red Pepper Powder Adulteration with Allura Red and Red Pepper Seeds Using Hyperspectral Imaging

**DOI:** 10.3390/foods12183471

**Published:** 2023-09-18

**Authors:** Jong-Jin Park, Jeong-Seok Cho, Gyuseok Lee, Dae-Yong Yun, Seul-Ki Park, Kee-Jai Park, Jeong-Ho Lim

**Affiliations:** 1Food Safety and Distribution Research Group, Korea Food Research Institute, Wanju-gun 55365, Republic of Korea; p.jongjin@kfri.re.kr (J.-J.P.); jscho@kfri.re.kr (J.-S.C.); ydy0401@kfri.re.kr (D.-Y.Y.); 2Smart Food Manufacturing Project Group, Korea Food Research Institute, Wanju-gun 55365, Republic of Korea; gslee@kfri.re.kr (G.L.); skpark@kfri.re.kr (S.-K.P.); jake@kfri.re.kr (K.-J.P.)

**Keywords:** shortwave infrared, red pepper, adulteration, classification, machine learning

## Abstract

This study used shortwave infrared (SWIR) technology to determine whether red pepper powder was artificially adulterated with Allura Red and red pepper seeds. First, the ratio of red pepper pericarp to seed was adjusted to 100:0 (P100), 75:25 (P75), 50:50 (P50), 25:75 (P25), or 0:100 (P0), and Allura Red was added to the red pepper pericarp/seed mixture at 0.05% (A), 0.1% (B), and 0.15% (C). The results of principal component analysis (PCA) using the L, a, and b values; hue angle; and chroma showed that the pure pericarp powder (P100) was not easily distinguished from some adulterated samples (P50A-C, P75A-C, and P100B,C). Adulterated red pepper powder was detected by applying machine learning techniques, including linear discriminant analysis (LDA), linear support vector machine (LSVM), and k-nearest neighbor (KNN), based on spectra obtained from SWIR (1,000–1,700 nm). Linear discriminant analysis determined adulteration with 100% accuracy when the samples were divided into four categories (acceptable, adulterated by Allura Red, adulterated by seeds, and adulterated by seeds and Allura Red). The application of SWIR technology and machine learning detects adulteration with Allura Red and seeds in red pepper powder.

## 1. Introduction

Red pepper (*Capsicum annum*) is extensively used as a spice in various cuisines and is commonly consumed as a powder. The government prohibits the addition of any substance to red pepper powder other than the red pepper seeds that it originally contains [1]. However, some individuals in the food supply chain intentionally add inexpensive materials, such as bran and saw dust, into red pepper powder to maximize their profits [2,3].

This study focused on red pepper seeds and Allura Red colorant as adulterants. Red pepper seeds are generally separated from red pepper to produce high-quality red pepper powder. Some traders add seeds that were not part of the original red pepper to increase the quantity of the product. Red colorants are added to mask the adulteration since the addition of seeds can reduce the red color of the red pepper powder [4,5]. Allura Red is a synthetic azo dye that is widely used in various food products and is also used in the adulteration of red pepper powder. The detection of Allura Red is necessary for consumer safety because it is absorbed into the gastrointestinal tract, enters the bloodstream, and interacts with proteins during transport and metabolism to cause disease [6].

Several analytical techniques (including high-performance liquid chromatography, capillary electrophoresis, and fluorescence spectroscopy) were used to determine the illegal addition of adulterants. However, these conventional destructive methods require skilled analysts and are time consuming since they require several procedures, such as sample extraction, pretreatment, and data analysis [6,7]. Therefore, nondestructive techniques are increasingly being adopted for onsite detection in the food supply chain, owing to their feasibility, cost effectiveness, and ease of operation [8].

Various imaging and spectroscopy techniques, such as RGB imaging and near-infrared (NIR) spectroscopy, were applied to evaluate food quality and safety. RGB imaging provides spatial information, and NIR spectroscopy provides spectral and multi-constituent information [9]. Meanwhile, hyperspectral imaging (HSI) employs a combination of conventional imaging and spectroscopy and can extract spatial-, spectral-, and multi-constituent information by detecting spectral changes in broadband light after interaction with a sample. Additionally, HSI shows sensitivity to minor components [9,10]. Therefore, HSI was recently used to detect wheat bran, rice bran, corn flour, and rosin powder in Sichuan peppers [11]. In addition, it was used to identify red pepper powder contaminated by Sudan dye [5].

A hypercube gathers three-dimensional blocks of data consisting of two spatial dimensions and one spectral dimension by the HSI system [12]. It consists of relevant and nonspecific background information; thus, extraction of useful information from high-dimensional hyperspectral data is essential to apply HSI techniques to the food industry [13]. Machine learning is regarded as an efficient tool to analyze complex datasets owing to its self-learning and adaptive capabilities and its strong fault tolerance [14]. Therefore, HSI combined with machine learning is widely utilized for quality evaluation of agricultural products [15], detection of specific adulterants [16], and quantification of food components [17].

Nevertheless, no studies have focused on the simultaneous detection of Allura Red and red pepper seeds in red pepper powder using HSI and machine learning. Therefore, this study aimed to establish a machine learning classification model that can identify red pepper powder contaminated with Allura Red and seed adulteration based on HSI spectra.

## 2. Materials and Methods

### 2.1. Materials

Red pepper pericarp powder (without seeds) and seeds were purchased from Red Pepper Mill Cheong-ju Shop (Seoul, Republic of Korea) and Danong Food (Dangjin-si, Republic of Korea). The ground samples were sieved through 20- and 50-mesh sieves. The powder that passed through a 20-mesh sieve, but was trapped on a 50-mesh sieve (average 35 mesh) was used for the experiment. Allura Red (FD&C Red No 40 powder; Sensient Colors LLC, St. Louis, MO, USA) was used as the coloring agent.

### 2.2. Sample Preparation

The pericarp and seed of red pepper were blended in different weight ratios (pericarp: seed = 100:0, 75:25, 50:50, 25:75, and 0:100) for the samples adulterated without the dye. The red pepper pericarp, seed, and mixture of red pepper pericarp/seed (5 g) were mixed with 10 mL of aqueous Allura Red solution (0.025% *w*/*v*, 0.05% *w*/*v*, and 0.075% *w*/*v*) for the dye-adulterated samples. The mixture was homogenized using a blender for 1 min and dried at 60 °C for 24 h. The composition of each group is presented in Table 1. The prepared sample (5 g) was placed in a petri dish (50 × 15 mm, SPL Life Sciences Co., Ltd., Pocheon-si, Republic of Korea) and flattened with a presser before the experiments (approximately 0.5 cm sample height). In this way, 30 samples were prepared for each group. The samples were weighed with an accuracy of within 1% deviation from the intended weight.

### 2.3. Color Measurement

The L*, a*, and b* values were obtained using a spectrophotometer (CM-700d; Konica Minolta Optics Inc., Tokyo, Japan), and the measurements were repeated 10 times. The hue angle and chroma were calculated as follows:$$Hue\;angle={tan}^{-1}\frac{b*}{a*}$$
$$Chroma=\sqrt{{\left(a*\right)}^{2}+{\left(b*\right)}^{2}}$$

Tukey’s test (*p* < 0.05) was performed for statistical analysis using SPSS Statistics 20 (SPSS Inc., Chicago, IL, USA).

### 2.4. Determination of Capsaicin and Dihydrocapsaicin Content

The capsaicin and dihydrocapsaicin content were determined using the method of Kim et al. [18], with some modifications. The sample (2 g) was mixed with 80 mL of 100% ethanol and extracted at 85 °C for 3 h using a reflux extractor (FTCOD-6; Chang Shin Science, Changwon, Republic of Korea). The extract was filtered through filter paper (No. 1, Whatman Co., Maidstone, UK) and then adjusted to 100 mL in a 100 mL volumetric flask. The solution was filtered using a 0.2 μm syringe filter (D2520; Echrome Science, Daegu, Republic of Korea), and the filtrate was used for analysis.

The capsaicin and dihydrocapsaicin content were determined using an ultra-high-performance liquid chromatography system, with a fluorescence detector (LaChromUltra L-2000U Series; Hitachi High-Technologies Corp., Tokyo, Japan) consisting of LaChrom Ultra C18 (2 µm, 2 mm I.D x 50 mm) (Ex = 280, Em = 325 nm). Chromatographic separation was performed using isocratic elution, with a mobile phase consisting of 1% acetic acid and acetonitrile (60:40 *v*/*v*). The flow rate, injection volume, acquisition time, and column temperature were 0.6 mL/min, 2 µL, 6 min, and 25 °C, respectively. The experiments were conducted in triplicate, and the results were expressed as mg/kg.

### 2.5. Data Collection

A hyperspectral system was custom designed and comprised a SWIR camera (PA320F300TCL, OZRAY, Gwangmyeong, Republic of Korea), spectrograph (N17E, Specim, Oulu, Finland), vision dome light (VTDL550*240, Vision Technology, Cheonan-si, Republic of Korea) with six halogen lamps (150W power), and sample linear stage (FBL80E1400, FUYU, Chengdu, China). An optical module (shortwave infrared (SWIR) camera, spectrograph, and vision dome light) of the SWIR system was fixed 460 mm above the sample, and SWIR spectral images (hypercube) were acquired by line-scan mode, while moving constant velocity motion by 275 mm/s using a sample linear stage.

The SWIR spectral images were recorded from 900 to 1700 nm, and the reflectance intensities of images was measured at averaged 3.45 nm intervals. The resolution of the SWIR spectral image was 320 (w) × 256 pixels (l), and the spectral band had 256 channels. However, the first 20 bands and the last 16 bands were deleted to remove noisy signals; thus, 220 spectral bands were used for analysis.

The frame rate was 260 fps, and the exposure time was 3.8 ms/line during spectrum acquisition. Thirty hypercube datasets were collected for each group. Region of interest (ROI) spectral extraction was performed using the principal component analysis (PCA)-based ROI function in Prediktera Breeze software (Prediktera AB, Umea, Sweden). The mean spectra of each sample were used for data analysis.

### 2.6. Data Analysis for Classification

The 600 images were randomly divided into training (70%, 21 samples from each group, 420 samples in total) for principal component analysis (PCA) using Evince software (Prediktera AB, Umea, Sweden).

Machine learning classification was performed using the classification learner in MATLAB R2022b software (MathWorks, Inc., Natick, MA, USA). The training of machine learning implemented k-fold cross-validation to enhance the reliability of the result. The k-fold cross-validation involved randomly dividing the entire dataset into k-subsets of equal sizes. One of these k-subsets for each iteration served as the validation data, while the remaining subsets were employed for training purposes [19]. A total of 420 samples (70% of the dataset) was allocated for training, while the remaining 30% (180 samples) was reserved as the test set. Additionally, a fivefold cross-validation was employed.

The feature selection algorithm (minimum redundancy maximal relevance (MRMR)) was performed in MATLAB R2022b software. The wavelength bands with high importance scores were identified by MRMR, followed by construction of the optimal subset for each machine learning classifier by wavelength selection. Three machine learning classifiers were used: linear discriminant analysis (LDA), linear support vector machine (LSVM), and k-nearest neighbor (KNN). True-positive (TP), true-negative (TN), false-positive (FP), and false-negative (FN) rates were used to calculate the model performance indicators (accuracy, recall, specificity, precision, and F1-score) with the following equations:$$Accuracy=\frac{TP+TN}{TP+TN+FP+FN}\times 100$$
$$Recall=\frac{TP}{TP+FN}\times 100$$
$$Precision=\frac{TP}{TP+FP}\times 100$$
$$F1-score=2\times \frac{Recall\times Precision}{Recall+Precision}\times 100$$

## 3. Results

### 3.1. Color Values of Red Pepper Powder in Relation to Different Pericarp and Seed Ratios and Allura Red Concentrations

The color values of red pepper powder in relation to the pericarp/seed ratio and Allura Red concentration are shown in Table 2. The L* values of samples containing only seed (P0) and a pericarp/seed mixture (P25, P50, and P75) decreased with increasing Allura Red concentrations, whereas those of pure pericarp powder (P100) increased after the addition of Allura Red. The a* value increased with increasing Allura Red concentrations. However, the magnitude of the increase in the a* value decreased as the pericarp ratio increased. The addition of Allura Red increased the b* values of P75 and P100, the hue angle of P100, and the chroma values of P25, P50, P75, and P100. However, the addition of Allura Red decreased the b* values of P0 and P25; the hue angles of P0, P25, and P50; and the chroma of P0.

PCA was performed using the L*, a*, b*, hue angle, and chroma values. Color is considered the most crucial quality parameter of red pepper powder [20]. However, PCA showed that it was difficult to distinguish some groups (P50A-C, P75, P75A-C, and P100B-C) from P100 using color values as an indicator (Figure 1). This result highlights the limitations of relying solely on color values for onsite detection to assess food quality in the food supply chain. One significant challenge is the difficulty in distinguishing adulterated products when the addition of colorants results in color values similar to those of authentic products. From this perspective, HSI, especially SWIR-based HSI, is a useful alternative to overcome the above limitations since it provides detailed information about several compounds in food [21]. Therefore, this study used SWIR technology to identify adulterated red pepper powder.

### 3.2. Hyperspectral Information and PCA Results in Relation to the Pericarp and Seed Ratio and the Presence of Allura Red

The spectral data based on the pericarp/seed mixture ratio showed that the reflectance in the 1500–1700 nm range decreased, but the reflectance in the 1000–1350 nm range increased as the pericarp percentage increased (Figure 2a). The spectral data of the pericarp/seed mixture without Allura Red (P0-100) were used for PCA. The score plot of the first two principal components (PCs) (explaining 99.3% of the variation) showed that the groups were well separated based on the peel/seed ratio (Figure 2b). Principal component loadings revealed the contributions of the wavelengths, and the valleys and peaks of the loading plot could be considered effective wavelengths [22]. The loading plot indicated that the wavelength at 1000–1165 nm and the pericarp ratio were positively correlated, while the wavelength at 1470–1690 nm and the pericarp ratio were negatively correlated (Figure 2c).

The capsaicin and dihydrocapsaicin contents in the pericarp were over 20 times higher than those in the seeds (Table 3). Thus, the capsaicin content increased as the pericarp percentage increased. Capsaicin (C_18_H_27_NO_3_) is an inherent component of red peppers, and many previous studies reported that capsaicin content affects the NIR spectra of red pepper powder [23,24]. The characteristic wavelengths of chili pepper capsaicin are 1035 nm (the second overtone of N–H), 1150–1210 nm (the second overtone absorption of C–H stretching), 1300–1500 nm (the second combination regions of C–H stretching), and 1530 nm (the first overtone of N–H) [25]. The second overtone region of C–H bonds (1120–1200 nm) were assigned to the capsaicin and dihydrocapsaicin content since this region contained at least 18 C–H bonds [26].

The score plot explained 98.8% of the variation when PCA was performed using a spectral dataset that included pure pericarp (P100), pericarp adulterated with 0.15% Allura Red (P100C), and Allura Red (R) (Figure 2e). R exhibits the highest reflectance over the entire wavelength range, followed by P100C and P100 (Figure 2d). The loading plot suggested that 1205 nm and 1559-1618 nm were the key wavelengths indicating the presence of Allura Red (Figure 2f). Kamil, Mohamed, and Shaheen [27] reported that Allura Red showed specific function groups, including NH stretching, C-N=N-C, stretching C=C aromatic, and CH_2_ bending. Therefore, the increase in the reflectance of red pepper powder adulterated with Allura Red may have been caused by these functional groups of Allura Red.

PC1 and PC2 explained 87.7% and 11.5% of the variation across the samples, respectively, when all datasets were subjected to PCA. The cumulative variance contributions of PC1 and PC2 were 99.3% (Figure 3a). P100, P75, and P50 were evidently separated from the samples adulterated with Allura Red. However, the samples adulterated with Allura Red overlapped with P25 and P0, and the adulterated samples could not be clearly separated based on the Allura Red concentration.

The classification of the pericarp/seed ratio in the Allura Red-contaminated sample was determined by PC1, and wavelengths at 1153, 1206, and 1460 nm predominantly influenced this discrimination. In contrast, the pericarp/seed ratio of pepper powder without Allura Red was discriminated by PC2, and the dominant wavelengths in the loading plot were 1103 nm and 1508-1622 nm, which is consistent with the loading plot results in Figure 2c. Confining these spectral characteristics solely to Allura Red might be premature considering the presence of numerous other red azo dyes. Nevertheless, the PCA results suggest the possibility of SWIR to identify red azo dye and seed impurities in red pepper powder. Therefore, a machine learning method was applied for effective classification.

### 3.3. Classification of Adulterated Red Pepper Powder by the Pericarp Ratio and Allura Red Concentration

Three machine learning algorithms (LDA, LSVM, and KNN) were employed to train the model for classifying red pepper powder adulterated with different pericarp ratios and Allura Red concentrations (20 categories).

Prior to applying machine learning classification, the optimal wavelength was selected by the minimum redundancy maximal relevance (MRMR) algorithm. MRMR aimed to choose a feature subset that maximizes dependence and relevance, while minimizing redundancy from the raw feature set [28].

Hyperspectral image data contain a vast number of spectral bands, resulting in an overabundance of information. Wavelength selection is an important step for efficient data processing considering the limitations of computer hardware. Moreover, wavelength selection is beneficial to improve model performance by eliminating irrelevant variables [29]. The top five wavelengths with high importance scores from MRMR were 1371.6, 1663.2, 1072.8, 1514.8, and 1141.5 nm (Figure 4a). The accuracy of classification models was compared based on the number of features to construct a suitable feature subset for each classification model (Figure 4b). Accuracy was defined as the ratio of correctly predicted samples to the total number of samples. The model accuracy increased as the number of features increased. However, the model accuracy no longer increased when the number of features crossed a certain threshold. The number of features was selected based on classifier accuracy as a function of the number of features, ensuring that the model accuracy for the feature model set deviated by less than 3% from the maximum accuracy. The selected feature numbers were 35, 20, and 20 for LDA, LSVM, and KNN, respectively.

The accuracy of the three algorithms was 76.9–100.0% in the training set and 80.6–98.9% in the testing set (Table 4). Overall, the results did not substantially differ between the training and testing sets. This indicated that the developed models were robust and did not show overfitting. Among these models, LDA exhibited the highest accuracy, followed by LSVM and KNN.

In addition to accuracy, other evaluation indices, such as recall, precision, and F1_score, were used to evaluate model performance. Recall represents the ratio of correctly predicted positive samples to the total number of positive samples, whereas precision represents the ratio of correctly predicted positive samples to the total number of positive samples predicted. Meanwhile, the F1_score is a weighted average of precision and recall [30]. The test set accuracy in LDA was 98.9%, recall was 97.8%, precision was 98.0%, and the F1-score was 97.8%. This indicated that LDA showed good performance in classifying different pericarp and seed ratios (Table 4). There was some misclassification of the Allura Red concentration at P75 and P100; however, LDA clearly determined the presence of seeds and Allura Red (Figure 5). LDA is regarded as a robust classifier, with an accuracy above 90% in studies on classification using HSI [13]. The superior performance of LDA in comparison with other machine learning methods was observed in classification studies of coffee bean contamination [31] and the infection of brown rice [32]. This is likely because LDA maximizes the rate between class variance and the interior of the class variance in any particular dataset to ensure the greatest separability [33].

### 3.4. Classification of Adulterated Red Pepper Powder by Adulterant Types

In onsite assessments, identification of the presence of an adulterant may be more important than quantification of each adulterant. Therefore, the samples were divided into four categories: acceptable (P100 and P75), adulterated with Allura Red (Ad-R; P100A-C and P75A-C), adulterated with seeds (Ad-S; P0-50), and adulterated with seeds and Allura Red (Ad-SR; P0A-C, P25A-C, P50A-C, and P75A-C). P75 was included in the acceptable group, along with P100, since previous studies reported that the red pepper seed content in dried red pepper powder varies from less than 10% to as much as 25%, depending on the country of origin [34].

The first five wavelengths with a high importance score from MRMR were 1153.3, 1660.5, 1552.5, 1020.0, and 1375.6 nm (Figure 6a). The accuracy of LDA, LSVM, and KNN was 79.8–100%, 85.5–99.5%, and 93.1–99.3%, respectively, as the number of features varied from 3 to 220 spectral bands, (Figure 6b). The selected feature numbers for LDA, LSVM, and KNN were 35, 20, and 35, respectively. The accuracies of the classification models in the training and test sets were 99.0–100% and 94.4–100%, respectively (Table 5 and Figure 7). The 4-category classification model showed better classification performance than the 20-category model. A previous study also reported improved accuracy of classification models when the categories were appropriately adjusted [35]. LDA clearly classified four categories (100% accuracy), whereas LSVM misclassified some of the Ad-S and Ad-SR samples as acceptable, but its accuracy improved overall in comparison with the 20-category classification.

KNN poorly performed when adulterated red pepper powder was classified by different pericarp ratios and Allura Red concentrations (20 categories) (Table 4). However, the accuracy of KNN in the training set increased from 76.9% to 99.0% after adjusting the number of categories to four. The classification ability of KNN is based on the determination of sample-type attributes using a defined set of K neighbors, without relying on a function generated during training. Therefore, KNN had good classification ability only in the absence of overlap between classes [36]. Consequently, the improved accuracy of the KNN model can be attributed to the reduced overlap between classes resulting from category adjustment.

Phillips and Abdulla reported that when the amount of adulterant added to a product is small, the spectrum does not change, leading to misclassification [37]. In this study, the added concentration of Allura Red was small (0.05–0.15%), but SWIR and machine learning were able to identify adulteration with high probability. We believe that the combination of SWIR and machine learning technology is an effective tool for quality assurance of red pepper powder, considering the cost and time-intensive nature of conventional analytical methods used for detecting adulteration.

## 4. Conclusions

The color results obtained from the colorimeter (CIE L*, a*, and b* values; hue angle; and chroma) cannot reliably indicate the presence of Allura Red and red pepper seeds in red pepper powder. Therefore, this study developed a machine learning model based on HSI to detect adulteration in red pepper powder. LDA showed the highest accuracy, recall, precision, and F1_score after applying machine learning to a dataset of 20 groups categorized by pericarp/seed ratio and Allura Red concentration. This indicated the superiority of the LDA model for classifying adulterated red pepper powder. In addition, LDA, LSVM, and KNN effectively discriminated among the accepted and contaminated products when classifying the samples into four groups (acceptable, adulterated by Allura Red, adulterated by seeds, and adulterated by seeds and Allura Red).

This study focused on the adulteration of red pepper powder with Allura Red and red pepper seeds. It provided insights that SWIR combined with an appropriate machine learning classification method is a reliable tool to detect specific adulterants. Moreover, an effective quality-monitoring process for red pepper powder can be expected if the dataset classification and machine learning classification methods are selected according to the user’s specific purpose and the desired level of quality assessment. Furthermore, a greater number of sample sizes and types will improve the robustness of the model data.

## Figures and Tables

**Figure 1 foods-12-03471-f001:**
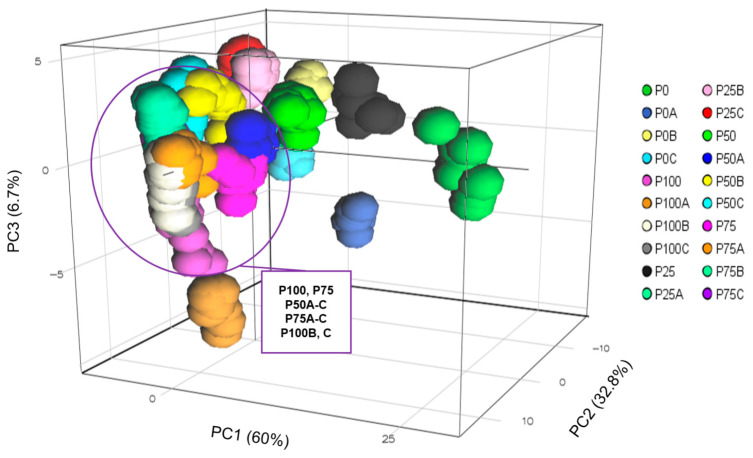
Principal component analysis (PCA) score plot for different pericarp/seed ratios and Allura Red concentrations. P0: pure seeds; P25: pepper powder consisting of 25% pericarp and 75% seeds; P50: pepper powder consisting of 50% pericarp and 50% seeds; P75: pepper powder consisting of 75% pericarp and 25% seeds; P100: pure pericarp; A, B, and C: Allura Red concentrations of 0.05%, 0.1%, and 0.15%, respectively, in the pericarp, seeds, and pericarp/seed mixture.

**Figure 2 foods-12-03471-f002:**
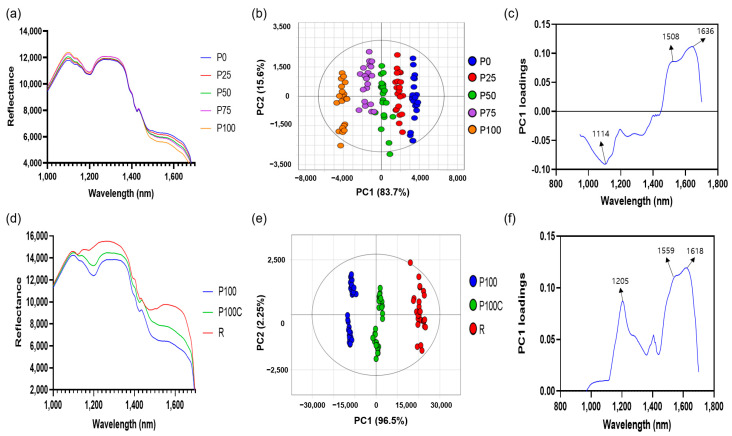
Shortwave infrared (SWIR) spectra (**a**), principal component analysis score (**b**), and loading plots (**c**) of red pepper powder with different pericarp and seed ratios (P0-P100) and SWIR spectra (**d**), principal component analysis score (**e**), and loading plots (**f**) of red pepper powder with Allura Red. P0: pure seeds; P25: pepper powder consisting of 25% pericarp and 75% seeds; P50: pepper powder consisting of 50% pericarp and 50% seeds; P75: pepper powder consisting of 75% pericarp and 25% seeds; P100: pure pericarp; P100C: pericarp adulterated with 0.15% Allura Red; R: Allura Red.

**Figure 3 foods-12-03471-f003:**
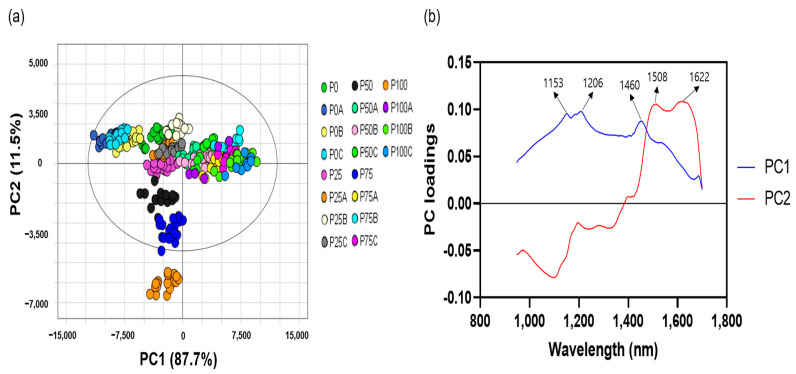
Principal component analysis score (**a**) and loading plots (**b**) of red pepper powder with different pericarp/seed ratios and Allura Red concentrations. P0: pure seeds; P25: pepper powder consisting of 25% pericarp and 75% seeds; P50: pepper powder consisting of 50% pericarp and 50% seeds; P75: pepper powder consisting of 75% pericarp and 25% seeds; P100: pure pericarp; A, B, and C: Allura Red concentrations in the pericarp, seeds, and pericarp/seed mixture at 0.05%, 0.1%, and 0.15%, respectively.

**Figure 4 foods-12-03471-f004:**
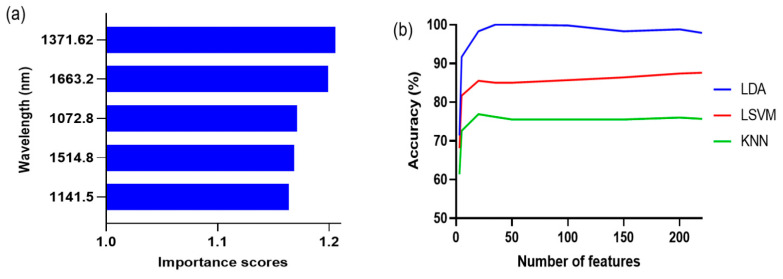
The first five wavelengths with high importance scores (**a**) and the classification accuracy by a selected number of features (**b**) from the minimum redundancy maximal relevance algorithm for 20-class classification. LDA: linear discriminant analysis; LVSM: linear support vector machine; KNN: k-nearest neighbors.

**Figure 5 foods-12-03471-f005:**
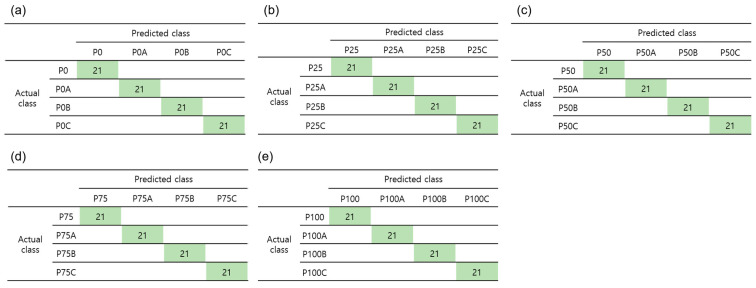
Confusion matrices of LDA in the training set for classification of red pepper powder by Allura Red concentration and pericarp/seed ratio (P0, P0A-C (**a**), P25, P25A-C (**b**), P50, P50A-C (**c**), P75, P75A-C (**d**), and P100, P100A-C (**e**), 20-class classification). P0: pure seeds; P25: pepper powder consisting of 25% pericarp and 75% seeds; P50: pepper powder consisting of 50% pericarp and 50% seeds; P75: pepper powder consisting of 75% pericarp and 25% seeds; P100: pure pericarp; A, B, and C: Allura Red concentrations in the pericarp, seeds, and pericarp/seed mixture at 0.05%, 0.1%, and 0.15%, respectively.

**Figure 6 foods-12-03471-f006:**
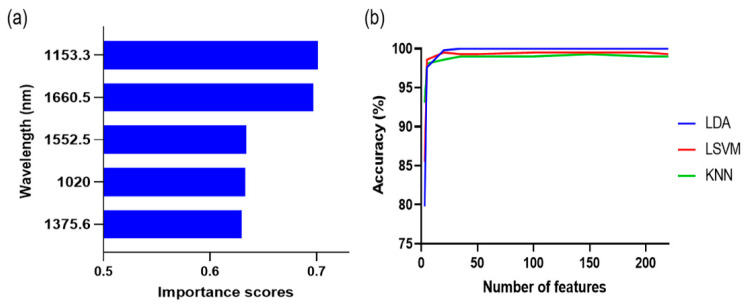
The first five wavelengths with high importance scores (**a**) and the classification accuracy by a selected number of features (**b**) from the minimum redundancy maximal relevance algorithm for 4-class classification. LDA: linear discriminant analysis; LSVM: linear support vector machine; KNN: k-nearest neighbors.

**Figure 7 foods-12-03471-f007:**
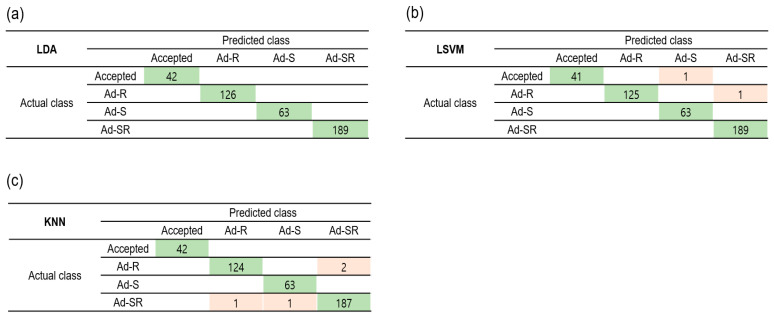
Confusion matrices of LDA (**a**), LSVM (**b**), and KNN (**c**) in the training set for classification of red pepper powder by adulterant types (4-class classification). Acceptable (P100 and P75), adulterated with Allura Red (Ad-R; P100A-C and P75A-C), adulterated with seeds (Ad-S; P0-50), and adulterated with seeds and Allura Red (Ad-SR; P0A-C, P25A-C, P50A-C, and P75A-C).

**Table 1 foods-12-03471-t001:** Appearance and composition of red pepper powder.

Group	Appearance	Pericarp (%)	Seed (%)	Allura Red (%)
P0	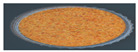	100.00	0.00	0.00
P0A	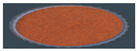	99.95	0.00	0.05
P0B	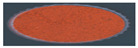	99.90	0.00	0.10
P0C	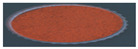	99.85	0.00	0.15
P25	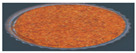	75.00	25.00	0.00
P25A	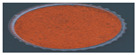	74.96	24.99	0.05
P25B	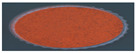	74.93	24.98	0.10
P25C	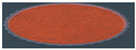	74.89	24.96	0.15
P50	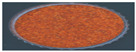	50.00	50.00	0.00
P50A	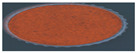	49.98	49.98	0.05
P50B	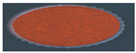	49.95	49.95	0.10
P50C	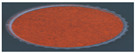	49.93	49.93	0.15
P75	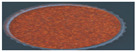	25.00	75.00	0.00
P75A	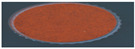	24.99	74.96	0.05
P75B	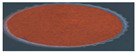	24.98	74.93	0.10
P75C	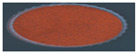	24.96	74.89	0.15
P100	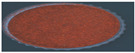	0.00	100.00	0.00
P100A	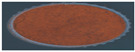	0.00	99.95	0.05
P100B	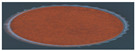	0.00	99.90	0.10
P100C	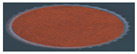	0.00	99.85	0.15

P0: pure seeds; P25: pepper powder consisting of 25% pericarp and 75% seeds; P50: pepper powder consisting of 50% pericarp and 50% seeds; P75: pepper powder consisting of 75% pericarp and 25% seeds; P100: pure pericarp; A, B, and C: Allura Red concentrations of 0.05%, 0.1%, and 0.15%, respectively, in the pericarp, seeds, and pericarp/seed mixture.

**Table 2 foods-12-03471-t002:** Color values for different pericarp/seed ratios and Allura Red concentrations.

	L*	a*	b*	Hue Angle	Chroma
P0	58.6 ± 0.7m	26.6 ± 1.1a	52.3 ± 1.5gh	63.1 ± 0.7n	58.7 ± 1.7de
P0A	43.2 ± 0.5k	31.1 ± 0.3b	34.1 ± 0.6a	47.7 ± 0.5c	46.2 ± 0.6a
P0B	41.1 ± 0.6j	38.9 ± 0.3h	34.9 ± 0.5a	41.8 ± 0.3a	52.3 ± 0.5c
P0C	37.3 ± 0.5gh	36.9 ± 0.6efg	34.1 ± 0.6a	42.7 ± 0.4a	50.2 ± 0.8b
P25	49.8 ± 1.0l	33.7 ± 0.8c	48.2 ± 1.3c	55.0 ± 1.0ij	58.8 ± 1.2de
P25A	41.0 ± 0.5j	37.1 ± 0.2fg	43.2 ± 1.1b	49.4 ± 0.7d	57.0 ± 0.9d
P25B	38.9 ± 0.5i	40.4 ± 0.2i	44.6 ± 0.6b	47.8 ± 0.4c	60.2 ± 0.5ef
P25C	37.7 ± 0.4h	41.9 ± 0.1j	43.3 ± 1.0b	46.0 ± 0.6b	60.2 ± 0.8ef
P50	43.5 ± 0.5k	36.2 ± 0.5de	49.1 ± 1.3cde	53.6 ± 0.8gh	61.0 ± 1.1f
P50A	38.5 ± 0.4i	37.1 ± 0.4fg	48.6 ± 1.1cd	52.7 ± 0.7fg	61.2 ± 0.9f
P50B	35.9 ± 0.5f	40.2 ± 0.5i	49.5 ± 1.9cde	50.9 ± 0.9e	63.8 ± 1.8g
P50C	33.4 ± 1.0d	41.4 ± 0.5j	50.5 ± 1.0ef	50.7 ± 0.5e	65.3 ± 1.0gh
P75	36.8 ± 0.6g	36.1 ± 0.5d	50.0 ± 1.3def	54.1 ± 0.6hi	61.6 ± 1.3f
P75A	34.1 ± 0.4e	37.6 ± 0.4g	55.8 ± 1.7i	56.0 ± 0.8kl	67.3 ± 1.5ij
P75B	33.2 ± 0.4cd	39.9 ± 0.5i	56.4 ± 0.7i	54.7 ± 0.3ij	69.1 ± 0.8j
P75C	32.5 ± 0.4bc	39.9 ± 0.4i	53.8 ± 1.0h	53.4 ± 0.3fgh	67.0 ± 1.0hi
P100	29.7 ± 0.6a	36.7 ± 1.1def	47.9 ± 2.4c	52.5 ± 0.6f	60.4 ± 2.5ef
P100A	32.9 ± 0.5cd	31.4 ± 0.4b	51.6 ± 1.4fg	58.7 ± 0.4m	60.4 ± 1.4ef
P100B	32.6 ± 0.7bcd	37.4 ± 0.5fg	55.8 ± 1.2i	56.2 ± 0.46l	67.2 ± 1.2hij
P100C	32.0 ± 0.7b	37.2 ± 0.5fg	53.6 ± 1.0h	55.2 ± 0.3jk	65.3 ± 1.1gh

Means in the same column with different letters are significantly different as per Tukey’s test (*p* < 0.05). P0: pure seeds; P25: pepper powder consisting of 25% pericarp and 75% seeds; P50: pepper powder consisting of 50% pericarp and 50% seeds; P75: pepper powder consisting of 75% pericarp and 25% seeds; P100: pure pericarp; A, B, and C: Allura Red concentrations of 0.05%, 0.1%, and 0.15%, respectively, in the pericarp, seeds, and pericarp/seed mixture.

**Table 3 foods-12-03471-t003:** Capsaicin content of pericarp and seed of red pepper.

	Pericarp	Seed
Capsaicin (mg/kg)	47.6 ± 0.2	2.0 ± 0.1
Dihydrocapsaicin (mg/kg)	32.0 ± 0.1	1.3 ± 0.1

**Table 4 foods-12-03471-t004:** Machine learning results for the classification of red pepper powder by different Allura Red concentrations and pericarp/seed ratio (20-class classification).

	Feature Number		Accuracy	Recall	Precision	F1_Score
Linear discriminant analysis (LDA)	35	Train	100.0	100.0	100.0	100.0
		Test	98.9	97.8	98.0	97.8
Linear support vector machine (LSVM)	20	Train	85.5	85.5	87.1	85.3
		Test	85.6	85.6	86.4	85.2
K-nearest neighbors (KNN)	20	Train	76.9	77.0	77.0	76.9
		Test	80.6	80.6	81.0	80.0

**Table 5 foods-12-03471-t005:** Machine learning results for classification of red pepper powder by adulterant types (4-class classification).

	Feature Number		Accuracy	Recall	Precision	F1_Score
Linear discriminant analysis (LDA)	35	Train	100.0	100.0	100.0	100.0
		Test	100.0	100.0	100.0	100.0
Linear support vector machine (LSVM)	20	Train	99.5	99.2	99.5	99.3
		Test	99.4	99.1	99.7	99.4
K-nearest neighbors (KNN)	35	Train	99.0	99.3	99.1	99.2
		Test	99.4	99.1	99.7	99.4

## Data Availability

The datasets generated for this study are available on request to the corresponding author.

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
