# Peer review of "Detection of Red Pepper Powder Adulteration with Allura Red and Red Pepper Seeds Using Hyperspectral Imaging"

_foods, 2023, doi:10.3390/foods12183471_

Round 1
Reviewer 1 Report
Dear Authors, here are my comments for the manuscript:
1. The authors could consider using a more typical 80/20 or 75/25 split for the training and test sets instead of 70/30. This may improve model performance.
2. Cross-validation should be performed during model development to evaluate the models more robustly and understand variance.
3. Using model interpretation methods like feature importance or partial dependence plots could give insight into which wavelengths are most important for classification. This type of analysis is missing and could strengthen the work.
4. The sample size of 30 per class seems small. A larger sample would likely make the models more robust. The authors should discuss the limitations imposed by the small sample size.
5. Data preprocessing like normalization, derivatives etc. was not performed. Evaluating different preprocessing techniques could impact model performance.
6. An independent test set completely separate from the training/validation sets should be used for final model evaluation. This is the best way to assess expected real-world performance.
7. Due to the small number of ingredients, there is likely high correlation between the major components like pericarp and seeds. This makes it difficult to differentiate the spectral response of each individual ingredient.
8. Using 3 principal components in the PCA color analysis instead of 2 may improve class separation. The authors should evaluate this.
9. Figures and tables should not be mixed together. Separating them makes it easier to evaluate the data.
I hope this helps.
Author Response
The authors would like to thank the reviewer for improving our paper. We have attached the author's note to reviewer.

Reviewer 2 Report
The manuscript entitled "Detection of red pepper powder adulteration with Allura Red and red pepper seeds using hyperspectral imaging" demonstrates the effectiveness of shortwave-infrared (SWIR) technology combined with machine learning techniques in accurately detecting adulteration with Allura Red and red pepper seeds in red pepper powder. The study highlights the successful separation of different adulteration levels and the identification of important wavelengths for determining the mixing ratio of pericarp/seed and the presence of Allura Red. Overall, this work presents a promising advancement in the field of food chemistry and quality control. However, I found the work need further improvement, especially in reporting the details of the study.
This research falls in with limited scope of adulterants. To be exact, the study focuses only on the adulteration of red pepper powder with Allura Red and red pepper seeds. It would be beneficial to expand the investigation to include other potential adulterants commonly found in the market, such as other food dyes or additives, to provide a more comprehensive analysis. While the manuscript highlights the effectiveness of SWIR technology and machine learning techniques, it could provide more insight into the practical implications of detecting the specific adulterants.
The manuscript could benefit from a comparison of the proposed SWIR-based approach with traditional or alternative methods currently used for adulteration detection in red pepper powder, such as portable NIR, FTIR or UV-Vis. As a matter of fact, the current SWIR did not take advantage of spatial resolution since only 1-D spectra is used for modelling. therefore, I would not consider it as hyperspectral imaging for this work. The authors should provide a clearer understanding of hyperspectral imaging, the advantages and limitations of it compared to established traditional techniques.
While the introduction part is a quite concise one, elaborate on the purposes of HSI in the food industry is needed. Rather than using the broad term "several purposes," at Line 58, I recommend to provide specific examples of how HSI combined with machine learning has been utilized for classification, detection, and quantification in the food industry. This would give readers a clearer understanding of the potential applications.
The parameters of each machine learning methods are crucial when comparing the performances between different methods and datasets. It seems no information given. That makes me wonder whether the authors have optimized these parameters when training the model. The training process should be justified. Typical parameters include:
Linear Discriminant Analysis (LDA):
Prior probabilities: The prior probabilities of each class in the dataset.
Shrinkage parameter: A regularization parameter that controls the degree of shrinkage applied to the covariance matrix estimation in case of small sample sizes.
Solver: The solver algorithm used to compute the LDA model (e.g., singular value decomposition, least squares solution, or eigenvalue decomposition).
Linear Support Vector Machine (LSVM):
C: The regularization parameter that controls the trade-off between maximizing the margin and minimizing the classification error.
Penalty: The penalty term used for regularization, such as "l1" for L1 regularization or "l2" for L2 regularization.
K-Nearest Neighbor (KNN):
Number of neighbors (k): The number of nearest neighbors to consider for classification. It determines the size of the neighborhood used to make predictions.
Distance metric: The distance metric used to calculate the distance between instances, such as Euclidean distance, Manhattan distance, or Minkowski distance.
Weighting scheme: The weighting scheme used to assign weights to the neighbors, such as "uniform" for equal weights or "distance" for inverse distance weighting based on the distance to the query point.
Other data processing parameters should be further clarified. How the ROI is selected? What is the image resolution of the raw data? How does the spectra combined to form the 1-D data for training? What is the final data size (samples or number of objects * data points or wavenumbers) for modelling?
Line 62: Clarify the term "hypercube": Explain what a hypercube refers to in the context of hyperspectral imaging, as it may not be familiar to all readers. Providing a brief explanation or definition would enhance understanding.
Line 84: Clarify the ratio, is it weight ratio?
Line 122: The authors only give the model of the SWIR camera. The setup of the whole acquisition device is not clear. How were the samples placed? How far away is the SWIR camera and lamp? What is the relevant position of camera, lamp and sample? Is there any method to avoid ambient lighting? Is there any specific method to avoid uneven distribution of powder?
Preprocessing in hyperspectral imaging plays vital roles in enhancing data quality, reducing noise, correcting for systematic errors, removing atmospheric interference, aligning images, and extracting meaningful features, etc. However, the authors did not mention anything related to hyperspectral imaging. It is recommended to add the relevant study. At least, the basic processing technique such as normalization or autoscaling should be evaluated.
Author Response
We appreciate your efforts in reviewing our paper. We have uploaded the review report with the corrections.

Reviewer 3 Report
The authors develop a method to detect the presence of a nonpermitted red dye (Allura Red) in red paprika and its adulteration with pericarp and seeds. For this purpose, they use hyperspectral imaging in the shortwave-infrared range (900-1700 nm). Obtained spectra were modeled using different chemometric techniques.
In general, the manuscript is well-written, but I encourage the authors to soften the conclusion (see remark no. 10). Concerning the number of raised queries, I recommend a major revision.
1) Section 2.2: it is not clear what is the meaning of symbols for different groups in Table 1. Please provide an explanation.
2) Section 2.2: how samples were weighted and what is the precision?
3) Section 2.4: what is the temperature of the column? If you are using a standard chromatographic method, please provide a reference. It was an isocratic separation?
4) Section 2.5: unfortunately, the description of how hyperspectral images were collected is imprecise. The authors use in their experiment the bush-broom hyperspectral system. What is the power of the illumination source(s)? Please state clearly if samples were moved under the camera or not. If yes, provide details of the movement speed. Frame rate is expressed in FPS or Hz. What do you mean by 10s? Do you really use 10s or Mili seconds for exposure? Please verify these measurement settings – with such measurement parameters sample imaging is ineffective.
5) Section 2.6: Explain how training and validation sets were constructed. It is necessary to guarantee a representative training set, namely, including all sources of expected variability. How was it assured?
6) Please provide justification for the selected considered in this study concentrations in prepared mixtures of paprika samples.
7) In section 2.2: Please clearly state how many samples in total were prepared. Did you prepare replicates? How many? How many spectra characterized each sample? The final discriminant models and PCA were carried out for mean spectra. How many samples were in the model and test sets?
8) Please add information what is the advantage of using hyperspectral imaging over classic near-infrared spectroscopy.
9) Figure 1a: revise the data provided as uncertainties. The quality (resolution) of the images must be improved to increase their clarity.
10) The authors use Allura Red as a dye. In section 3.2, specific bands are indicated. Since Allura Red is one of many azo dyes, I wonder about this approach’s specificity. I am afraid that concerning possible variability of the material and the large number of red azo dyes with very similar spectral characteristics, claiming the detection of Allura Red is premature. The authors construct the so-called hard classification rules, and in my opinion, such an approach is capable of verifying the existence of chemical differences between genuine and counterfeit samples but not indicating the type of red azo dye. Soft classification methods, constructed using, for instance, the SIMCA method, are appropriate when the number of groups (possible red nonpermitted dyes) is a priori unknown. I encourage the authors to soften the discussion and include their opinion in the revised manuscript.
11) The authors use three methods LDA, LSVM, and KNN. Why? Justify their selection.
12) Figure 4 – this figure is redundant because necessary information is provided in Table 3. Figures 5a-d are redundant. Please add a Table with the content of Figure 5d.
Author Response

(The authors gave the same response as above.)

Round 2
Reviewer 1 Report
Dear Authors,
Thank you for your revised manuscript. I am pleased to see that the work has improved significantly.
I concur with your results and appreciate the modifications you made to the presentation of the color data. I performed my own PCA models with the data you presented in Table 2. Specifically, I observed a slight separation of the samples, similar to your findings. However, I noticed that the models based on the SWIR data yielded even better outcomes.
I have one minor suggestion regarding Figure 1. To avoid potential confusion or misinterpretation, I recommend that you standardize the size of the score markers for all samples. Using different marker sizes without a statistical justification could lead to ambiguity.
Once again, thank you for the revisions you've made.
Author Response
The size of the score markers is standardized (Figure 1). Once again, we thank for your thoughtful suggestions and insights.
Reviewer 2 Report
All my comments made in the previous version were clearly responded and revision made accordingly. Thus, I agree to publish this work.
Author Response
Once again, we thank for your thoughtful suggestions and insights.
Reviewer 3 Report
Dear Authors,
Thank you for providing answers to my comments and preparing the revised manuscript. In my opinion, it is ready to be published.
Kind regards,
Author Response

(The authors gave the same response as above.)
